# Improved Electrical Characteristics of Field Effect Transistors with GeSeTe-Based Ovonic Threshold Switching Devices

**DOI:** 10.3390/ma16124315

**Published:** 2023-06-11

**Authors:** Su Yeon Lee, Hyun Kyu Seo, Se Yeon Jeong, Min Kyu Yang

**Affiliations:** Artificial Intelligence Convergence Research Laboratory, Sahmyook University, Seoul 01795, Republic of Korea; suyeonleel@naver.com (S.Y.L.); seohyunkyu0811@gmail.com (H.K.S.); seven77391@gmail.com (S.Y.J.)

**Keywords:** threshold switching, MOSFET, hyper-FET, sub-threshold slope, ovonic threshold switch, steep slope, logic circuit

## Abstract

Hyper-field effect transistors (hyper-FETs) are crucial in the development of low-power logic devices. With the increasing significance of power consumption and energy efficiency, conventional logic devices can no longer achieve the required performance and low-power operation. Next-generation logic devices are designed based on complementary metal-oxide-semiconductor circuits, and the subthreshold swing of existing metal-oxide semiconductor field effect transistors (MOSFETs) cannot be reduced below 60 mV/dec at room temperature owing to the thermionic carrier injection mechanism in the source region. Therefore, new devices must be developed to overcome these limitations. In this study, we present a novel threshold switch (TS) material, which can be applied to logic devices by employing ovonic threshold switch (OTS) materials, failure control of insulator–metal transition materials, and structural optimization. The proposed TS material is connected to a FET device to evaluate its performance. The results demonstrate that commercial transistors connected in series with GeSeTe-based OTS devices exhibit significantly lower subthreshold swing values, high on/off current ratios, and high durability of up to 10^8^.

## 1. Introduction

Research on ultralow-voltage beyond-complementary metal-oxide semiconductor (CMOS) technology, capable of reducing leakage power and driving voltage, is crucial for developing next-generation semiconductor technologies. To achieve ultra-low power, the field effect transistor (FET), a transistor with a low sub-threshold swing (SS), low leakage current, and a low operating voltage in a region far lower than 60 mV/dec at a limit of 300 K of “ln(10)·kT/q” must be developed to enable low-power computing in the future [1,2,3]. This is because the lower the SS value, the greater the change in current, even if the gate voltage is slightly adjusted, the lower the power consumption in the off state. However, the Boltzmann theory states that the basic limit of SS at room temperature is 60 mV/dec, which is a significant barrier to lowering the operating voltage of conventional metal oxide semiconductor field effect transistors (MOSFETs) [4,5,6,7,8,9]. Considering the unit of SS value, it means the size of gate voltage required for the drain current value to increase by 10 times. Consequently, several alternatives have been proposed, such as tunneling FET (TFET) technology and negative capacitance FET (NC-FET) technology, which implement tunneling between bands [10,11,12,13,14,15,16,17]. However, TFET technology faces technical drawbacks such as device structure, size, and low driving current, and NC-FETs face low-reliability problems such as the wake-up effect of ferroelectric oxides and defect generation. Therefore, hyper-FET is one of the technologies that can be implemented as an alternative. The hyper-FET operates similarly to the TFET and achieves a low SS by attaching VO_2_ [18,19,20,21], an insulator-to-metal transition (IMT) material whose SS is enhanced to less than 8 mV/dec compared to the source. Furthermore, it reduces the voltage required to generate a current. However, it is difficult for the hyper-FET to achieve a low leakage current and low operating bias conditions from the electrical properties of VO_2_ [22,23,24,25]. Various research groups have proposed replacing VO_2_ with an electrochemical metallization (ECM)-type threshold switching (TS) device with very high off-state resistance to solve this problem. The SS can be reduced to the level of 10 mV/dec by utilizing the approximate current increase during TS [26,27,28]. Recently, the ovonic threshold switch (OTS), which exhibits a fast-switching speed and low leakage current, has gained considerable research interest as a device that can replace the TS [29,30,31,32,33,34,35,36,37]. The OTS is primarily composed of amorphous chalcogenide-based non-metallic materials. It is a volatile memory that maintains a high-resistance state in an electrically inactive state and switches to an active state when a voltage above a certain value is applied to reduce the resistance. The OTS presents excellent power consumption efficiency and high speed since it can be switched even at low voltages. It is a high-performance device that has gained considerable attention in the field of artificial intelligence calculations. The OTS enables switching only when the current value is above a certain level, to overcome the theoretical limit of the MOSFET [38,39,40]. In this study, an OTS device with a steep turn-on/off slope, high durability, low operating voltage, and a fast-switching speed of 100 ns was developed. The OTS device with a W/GeSeTe(GST)/W structure was connected in series to the commercial transistor source part, and the proposed device reduced the swing value to 20 mV/dec.

## 2. Materials and Methods

Figure 1a presents a 3D diagram of the W (top electrode)/GeSeTe(GST)/W (bottom electrode) structure employed in this study. To fabricate the lower electrode W, a dry etching process using inductively coupled plasma reactive ion etching (ICP-RIE, LAT, Suwon, Korea) equipment was employed after the photolithography of the Si/SiO_2_/W wafers. The patterning of the lower electrode W was achieved by subjecting the wafers to ICP-RIE with a cell size of 16 μm^2^.

The ICP power was maintained at 300 W, while the substrate bias power was set to 20 W during the etching of the W layer. Ar and Cl_2_ gases at flow rates of 4 sccm (standard cubic centimeters per minute) and 30 sccm, respectively, were used in the dry-etching process. The temperature was controlled at 10 °C using a water-circulation cooling system to ensure optimal process conditions. The observed dry etching rate was approximately ~1.5 nm/s, indicating the effective removal of the material during the etching process. Subsequently, the W bottom electrode (BE) was washed with isopropyl alcohol and deionized water after removing the remaining photoresist (PR) with acetone. The GST layer was deposited using a sputtering device. The GeSeTe (35:50:15) and Te targets were deposited at a rate of 2 rpm through co-sputtering. A GeSeTe layer with a thickness of 50–60 nm was deposited via radiofrequency (RF) magnetron sputtering. The GeSeTe sputtered at 50 W and Te at 30 W power. The base pressure of the chamber was set to less than 4.0 × 10^−6^ Torr, and the working pressure was maintained at 2.0 × 10 Torr by maintaining the Ar (purity of 99.999%) flow. The upper W electrode was deposited with an OTS layer and then deposited and patterned using the lift-off method with a W target. A transmission electron microscope (TEM, JEOL, Akishima, Japan) analysis was conducted to analyze the detailed structure of the GeSeTe thin film. The samples used for TEM analysis were prepared using focused ion beam (FIB, crossbeam 540, Zeiss, Jena, Germany) techniques. Figure 1b presents a cross-sectional view of the W/GeSeTe(GST)/W structure, which highlights the layer arrangement and interfaces within the thin film. In addition, Figure 1c presents the results obtained through energy-dispersive X-ray spectroscopy (EDS, JEOL, Akishima, Japan), which depicts the elemental distribution of W, Ge, Se, and Te within the thin film. The EDS mapping clearly shows that each layer is precisely defined, indicating the accurate deposition and composition of the different materials.

## 3. Results

Figure 2a depicts the DC I-V curve, which is an electrical characteristic of the device. The V_th_ was within 1 V, and the on/off ratio was three orders of magnitude. Furthermore, the I_off_ of the proposed OTS device was less than 10 nA. The compliance current (CC) was set to 100 μA, and (Keithley 4200) electrical pulse-based measurements were performed using a function generator (AFG, Agilent 81110A, Beaverton, OR, USA) oscilloscope (MSOX3024T, Tektronix, Beaverton, OR, USA), and the resistance state was confirmed. The resistance programming and verification were performed simultaneously owing to the volatile characteristics of the OTS device. The AFG, the tested OTS element, and the OSC were connected in series. Figure 2b depicts the measurement of the delay time when the OTS was turned on using the oscilloscope. It can be observed that the GeSeTe-based OTS device is converted in less than 10 ns; the device operates quickly when switching since it does not involve atomic rearrangement or structural changes [34,35].

Figure 3a depicts the endurance performance of the OTS device. The volatile OTS is distinct from non-volatile memory such as resistive random-access memory (RRAM). A burst read scheme, which differs from the conventional approach of repeated write-and-read operations, was employed to evaluate its performance. After 10 to n numbers of pulses were applied to the OTS device cell the resistance of the OTS device was determined. This unique method enables a reduction in read damage when compared to conventional read schemes. We determined the resistance of the OTS cell by applying a specific number of pulses. This approach ensures accurate measurements while mitigating the potential for read-induced damage. Thus, the burst read scheme can be used to effectively assess the performance of an OTS device.

Figure 3b presents a graph comparing the results of the pulse-based current-voltage (PIV) measurements before and after conducting 1 × 10^12^ durability tests. This comparison verified the exceptional endurance performance of the OTS device, demonstrating its ability to maintain reliable performance over prolonged periods of usage.

Figure 3c depicts the change in the threshold voltage (V_th_) between 10 ns and 10 μs pulse width. The results demonstrate minimal variation in the V_th_ value, indicating the consistent and stable operation of the OTS device regardless of the time. This demonstrates the fast and reliable switching capabilities of the OTS device while maintaining consistent performance over a time range of up to 10 s. Figure 3d presents the drift measurements of the OTS device. For the OTS based on chalcogenide materials, drift phenomena may occur over time, potentially affecting the overall device performance. The drift value must remain below 20 mV/dec to ensure successful commercialization. Figure 3d depicts the measured drift value of the GeSeTe volatile memory, which was 18 mV/dec. These results confirm the stability and reliability of the OTS device, further enhancing its practical applications and commercial potential.

In summary, the comprehensive analysis presented in Figure 3 highlights the exceptional endurance, fast and consistent operation, and ability to maintain stable performance over extended periods, owing to which the OTS device is a highly promising candidate for next-generation semiconductor technology.

Figure 4a,b presents a simple schematic diagram of a commercial transistor connected in series with and without the GeSeTe-based OTS device. The OTS was connected to the source part of the transistor, and the ground was held in the BE part for measurement. Figure 4c presents the transfer curve of a transistor without an OTS device, and Figure 4d presents the characteristics of a transistor with an OTS device connected in series. The commercial transistor used in this study was VN2222, which is an n-type MOSFET with a maximum voltage V_DS_ of 60 V, maximum current I_D_ of 0.26 A, and a maximum power P_D_ of 0.3 W. In addition, the gate-source voltage (V_GS_) is ±20 V, gate voltage (V_GS_ (th)) is 2–4 V, and the channel resistance (R_DS_ (on)) is 18 Ω. The fixed drain voltages of V_th_ and 1V were applied during the I_DS_ V_GS_ sweep.

The power consumption of a transistor is divided into operating power consumption and standby power consumption. The operating voltage of the transistor and the standby current must be reduced simultaneously to achieve low power. Therefore, minimizing the SS value is crucial. The sub-threshold swing (SS) refers to the inverse of the transfer characteristics, the value obtained by taking semi-log of the (Id-Vg graph) drain current, divided by gate voltage. The sub-threshold swing (SS) is low in the sub-threshold voltage range because even a little reduction in voltage can easily control on/off, thereby reducing power consumption. The SS is low in the sub-threshold voltage range because even a little reduction in voltage can easily control on/off, thereby reducing power consumption. The SS value can be determined through Equation (1):(1)$$SS=\ln(10)\frac{kT}{q}\cdot n≅n\cdot 60\;\mathrm{mV}/\mathrm{dec},$$
(2)$$n=1+\frac{C_{d}}{C_{ox}}$$
where *C_d_* is the depletion layer capacitance and *C_ox_* is the gate oxide capacitance. At 300 K, the *SS* value is already 60 mV/dec, which can be determined by the equation. To reduce the *SS* value, we connected the OTS device to the transistor, and as a result, we were able to reduce the slope value. Comparing the SS values on the transmission curve shows that the *SS* values of the transistor-only devices (Figure 4b) are more than twice as different from those of the OTS devices (Figure 4c) connected in series. It succeeded in reducing the *SS* value to 20 mV/dec by connecting the transistor and the OTS element. A high on/off current ratio of more than 10^5^ was also observed. When compared under the same conditions, the leakage current is also reduced.

Figure 5a depicts the obtained output characteristic curve, which is essential for understanding the operational characteristics of transistors, allowing the identification of the regions of transistor operation, where current amplification and voltage attenuation occurred. An analysis of the output curve demonstrated that the current compliance level varied with the step of the gate voltage. Figure 5b illustrates the case where the gate voltage is set to 1.5 V, controlling the current compliance at 100 μA. Since the pulse operation must be shown to measure the endurance, we checked the characteristics of the output curve after setting the transistor + OTS device to 1.5 V gate voltage. As a result, sweep operations were shown, followed by reliability measurements in hyper-FET. This enabled us to observe the output curve under specific conditions. The I-V curve depicted in Figure 5b demonstrates sweep behavior, followed by the measurement of reliability in the hyper-FET. This analysis of the output curves helps in obtaining an understanding of the operational regions and characteristics of the transistor, which further helps in appropriately designing transistors and establishing optimal operating conditions. Furthermore, for low-power logic devices such as the hyper-FET, the output curve analysis enables the evaluation of the tradeoff between power consumption and performance.

In the previous section, the endurance of the device was measured by adjusting the gate voltage step in accordance with the output power curve. An endurance test was conducted, and the results are presented in Figure 6a. The measurement results confirmed that the device could withstand up to 10^8^ pulses without significant degradation, and the pulse width was measured to be 50 ns.

Subsequently, the pulse waveform was analyzed when the switching characteristics were verified by bonding the transistor and the GeSeTe-based OTS device in series, as shown in Figure 6b. It was determined that the gate delay operated within 20 ns, demonstrating the effectiveness of the device in switching applications. Overall, these findings provide valuable insights into the performance and reliability of the device, which may be useful for designing and optimizing similar devices in the future.

## 4. Conclusions

Research on ultralow-voltage beyond-CMOS technology, which can significantly reduce the leakage power and driving voltage, is becoming increasingly important for the development of next-generation semiconductor technologies. The SS of the existing MOSFETs cannot be reduced below 60 mV/dec at room temperature owing to the thermionic carrier injection mechanism in the source region.

This study presented a GeSeTe-based OTS device with the advantages of a fast-switching speed and low leakage current to overcome the SS limit of MOSFETs at room temperature. These OTS devices were connected in series with transistors to analyze the changes in the SS. The proposed GeSeTe-based OTS device exhibited excellent characteristics including a leakage current of less than 10 nA, steep on/off slope of over 10^3^, and fast switching speed. Additionally, these devices demonstrated high endurance and operated stably for up to 10^12^ cycles.

Consequently, the hyper-FET formed by connecting the OTS device with transistors presented an improved performance when compared with standalone transistors and confirmed stable endurance for up to 10^8^ cycles. The OTS device connected in series with the transistors presented an SS of less than 20 mV/dec and a high I_ON_/I_OFF_ ratio of over 10^5^. We have identified the potential to overcome the physical limitations of SS at 60 mV/dec and improve both performance and energy efficiency. The development of SS in FET technology is expected to have a significant impact on the energy-efficient modern society. This result not only improves the performance of the MOSFET but also. As more and more applications are required to implement features such as IoT and wearable processors at ultra-low power, these technologies can be used effectively in circuit systems that require ultra-low power. This will help improve technology and price competitiveness. It can also reduce the power consumption of existing electronic components that are high in power consumption, improving operational life. It is expected to contribute to the development of semiconductor technology in the future and promote energy conservation and the development of high-performance electronic devices.

## Figures and Tables

**Figure 1 materials-16-04315-f001:**
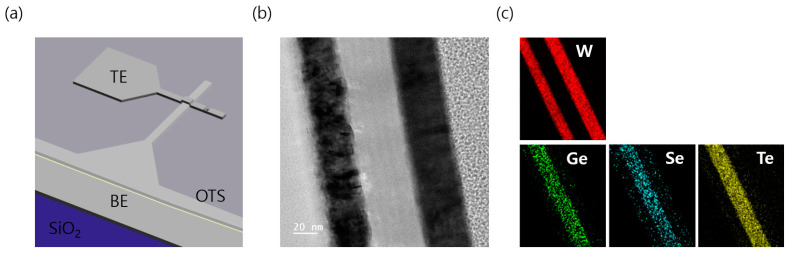
(**a**) 3D structure of the W/GeSeTe/W, (**b**) Cross-sectional view of the W/GeSeTe(GST)/W, (**c**) Energy Dispersive X-ray Spectroscopy (EDS) of W/GeSeTe/W.

**Figure 2 materials-16-04315-f002:**
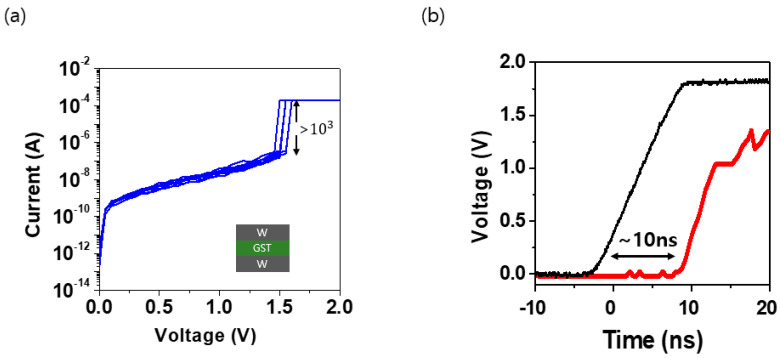
(**a**) DC I–V curves of W/GeSeTe/W device, (**b**) turn–on voltage measured using an oscilloscope.

**Figure 3 materials-16-04315-f003:**
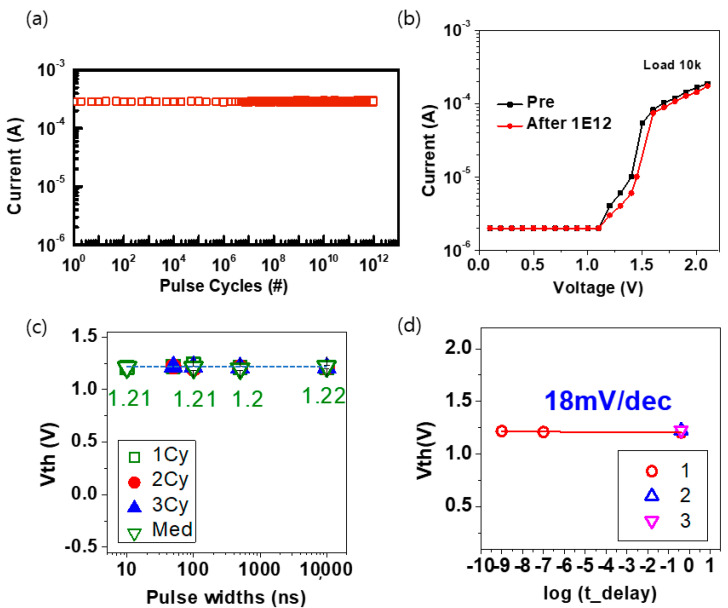
(**a**) The endurance characteristics of the OTS device, (**b**) comparison of the pulse–based current-voltage (PIV) results before and after the endurance test, (**c**) threshold voltage (V_th_) endurance with different pulse width, and (**d**) V_th_ shift with a delay time on a log scale.

**Figure 4 materials-16-04315-f004:**
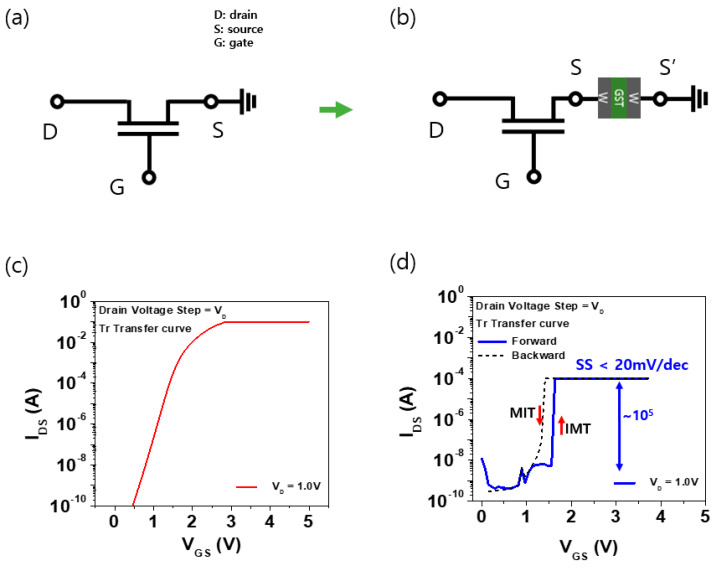
(**a**) Schematics of transistors without the OTS device, (**b**) schematics of transistors with the device, and (**c**) IDS–VGS characteristics of a transistor without the GeSeTe–based OTS device, (**d**) with the device.

**Figure 5 materials-16-04315-f005:**
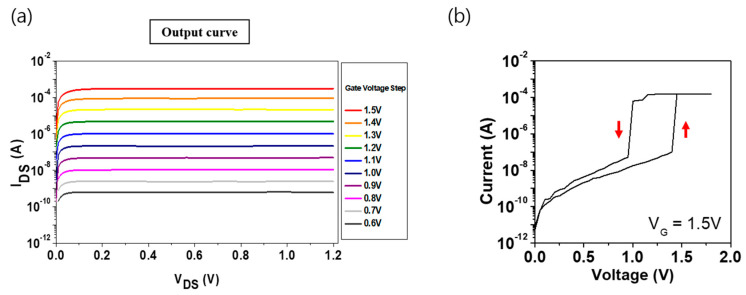
(**a**) Output curve of transistor, (**b**) output curve of transistor + GeSeTe–based OTS device.

**Figure 6 materials-16-04315-f006:**
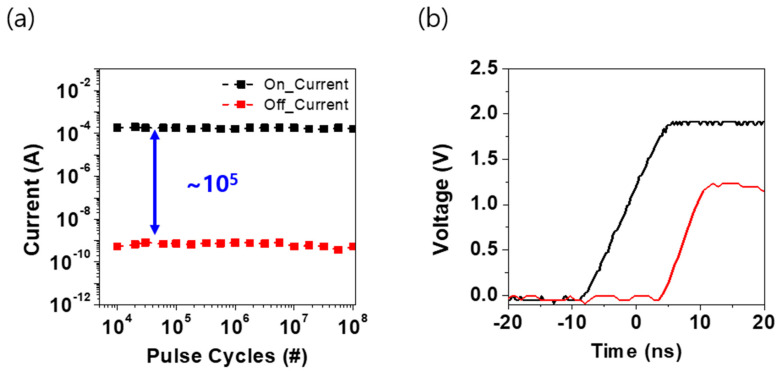
(**a**) AC endurance up to 10^8^ cycles (**b**) delay–time during pulse application.

## Data Availability

Not applicable.

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
