# Peer review of "Improved Electrical Characteristics of Field Effect Transistors with GeSeTe-Based Ovonic Threshold Switching Devices"

_materials, 2023, doi:10.3390/ma16124315_

Round 1

Reviewer 1 Report

S. Y. Lee et al. have proposed a TS material to work in conjunction with a commercial FET device in order to evaluate its performance. The study has produced interesting results, and the paper is well-organized and informative. Below are the reviewer's comments:

1) Figure 4 contains four sketches. Please revise the figure caption accordingly.

2) In Figure 4c, the proposed hybrid device exhibits a form of hysteresis. Please provide a comment on this observation.

3) Upon inspecting the same figure, it can be observed that there is a decrease in the on-state current at the expense of the subthermionic subthreshold swing. Please provide a comment on this significant observation.

4) It is recommended to include some explanation regarding the scaling capability of this hybrid device.

Considering the comments mentioned above, a minor revision is recommended. Good luck!

N/A

Reviewer 2 Report

The manuscript presents results on obtaining improved characteristics of field-effect transistors integrated with GeSeTe-Based OTS devices. The manuscript lacks significant academic rigor and needs significant improvement in the data presentations and analysis. The comments below show what must be improved to become a publishable quality.

1. lines 36-37, the statement "When the voltage is increased by ten times, the current increases by ten times accordingly." is not substantiated and it is not clear which currents (source-drain, leakage) and voltages are meant here. The context implies source-drain current vs gate voltage, but then the statement is not true. 

2. line 75, a "W" is missing in the structure definition W/GeSeTe/,

3. line 84, "BE" acronym is not properly introduced.

4. line 92, "DCI-V" should be "DC I-V" with the space added in the middle.

5. line 108, the abbreviation for "PIV" is not introduced at this stage of the manuscript.

6. lines 109-110, Vth is used with subscript the and without, Please use one way consistently through the manuscript.

7. In Figure 4 (a), on the right-hand side, it would be beneficial to show the connection of S and S' to the W, but not to GST as depicted in the diagram.

8. In Figure 4 (c), several curves are merged and it is difficult to discriminate the measured data. Besides, there is a big difference in shown curves, so more measurements performed will help to support the claim in this manuscript. The data is not discussed in much detail. The lack of proper captions and discussion makes the data unverifiable. Statistical analysis for several samples is not demonstrated making the scientific claims in this paper less substantiated. Statistics for at least several samples must be presented.

9. lines 145-146, the Figure 4 caption is missing and instead, the Fig. 3 one is duplicated by mistake.

10. lines 161-162, the Figure 5 caption is missing and instead, the Fig. 3 one is duplicated by mistake. It is difficult to understand what Figure 5(b) results are. Figure 5 (a) has a limited scope of output characteristics where the gate voltages in the range of 1.2-1.5 V are used. There is not enough data to show a full understanding of the new device features.

Proof reading would be recommneded

Reviewer 3 Report

This paper demonstrated FET operation of a unique device involving a bistable chalcogenide material. The results seem to be basically worth publishing, but I feel that the manuscript was rather poorly prepared.

I think that the word “OTS” is not generally known. It should be spelled out in the title. The word “Ovonic” should be explained in the text.

Figs.1 (b) and (c) are a part of the results, and thus should be given in the third section.

The SEM and EDS results should be given first, before the electrical data.

I did not understand the oscilloscope data Fig.1(c) and Fig.6(b). What is the red curve ?

The captions of Figs.4 and 5 are obviously wrong. The label of the vertical scale of Fig.6(a) is also wrong.

Figs.5 (a) and (b) do not seem consistent. At V=1.2V, for example, the device could be either on or off, according to Fig.(b). It should be discussed whether the multiple curves in Fig.5(b) leads to unstable behaviors of the device.

If the current showed hysteresis, arrows should be used to distinguish positive and negative sweeps (Fig.4(c), Fig.5(b)?).

The text should be checked, edited overall. There are obvious errors, e.g.,

“after it was subjected to a series of pulses ranging from 10 to n”

“the change in the threshold voltage (Vth) between 10 μs and 10 μs pulse width”

Round 2

Reviewer 2 Report

The resubmission takes into consideration most of the reviewer's comments making the manuscript of publishable quality.

Minor proof-reading would be adavntegeous

Reviewer 3 Report

I still did not understand the oscilloscope data Fig.1(c) and Fig.6(b). Which voltage was measured for the black and red curves ? Please describe in the text.

Why there are two curves in Fig.4(c), Fig.5(b) ? (The results were not reproducible ?)

The following errors were not corrected. The text should be checked overall.

“after it was subjected to a series of pulses ranging from 10 to n”

“the change in the threshold voltage (Vth) between 10 μs and 10 μs pulse width”

none
